# The Influence of Various Hydration Strategies (Isotonic, Water, and No Hydration) on Hematological Indices, Plasma Volume, and Lactate Concentration in Young Men during Prolonged Cycling in Elevated Ambient Temperatures

**DOI:** 10.3390/biology12050687

**Published:** 2023-05-07

**Authors:** Tomasz Pałka, Piotr Michał Koteja, Łukasz Tota, Łukasz Rydzik, Marta Kopańska, Izabela Kaczorowska, Norollah Javdaneh, Wioletta Mikulakova, Hubert Wolski, Tadeusz Ambroży

**Affiliations:** 1Department of Physiology and Biochemistry, Faculty of Physical Education and Sport, University of Physical Education, 31-571 Kraków, Poland; 2Institute of Sports Sciences, University of Physical Education, 31-571 Kraków, Poland; 3Department of Pathophysiology, Institute of Medical Sciences, Medical College of Rzeszów University, 35-959 Rzeszów, Poland; 4Doctoral Studies Institute of Biomedical Sciences, University of Physical Education, 31-571 Kraków, Poland; 5Department of Biomechanics and Sports Injuries, Kharazmi University of Tehran, Tehran 14911-15719, Iran; 6Department of Physiotherapy, Faculty of Health Care, University of Presov, 080 01 Presov, Slovakia; 7Medical Institute, Podhale State Vocational College, 34-400 Nowy Targ, Poland; 8Department of Perinatology and Women’s Diseases, Poznań University of Medical Sciences, 61-701 Poznań, Poland

**Keywords:** hydration, hyperthermia, dehydration, hematological markers, young men

## Abstract

**Simple Summary:**

Engaging in physical exertion in an elevated temperature environment leads to water–electrolyte disturbances as well as to the disruption of thermal balance and internal homeostasis, which enhances the physiological–biochemical changes associated with the physical work. In order to maintain the body’s exercise capacity, it is essential not only to hydrate the body but also to provide it with an adequate amount and quality of carbohydrates. Disturbances of thermal and fluid balance homeostasis pose a health risk, exercise performance may be reduced in the heat, and athletes are therefore likely to attempt to reduce this performance loss. The aim of this study was to evaluate the effect of different hydration strategies (isotonic, water, and no hydration) on young men’s hematological indicators, plasma volume, and blood lactate levels during prolonged physical exercise in a high ambient temperature. Water and isotonic hydration strategies allow for a better maintenance of water–electrolyte homeostasis during physical activity in a high-temperature environment, but consumption of isotonic drinks has a greater effect on the hydration of extracellular spaces with minimal changes in hematological indices.

**Abstract:**

Background: Increased internal body temperature during dehydration can be accompanied by water–electrolyte imbalances, higher levels of lactate during and after physical exertion, and changes in blood volume. Adequate hydration with carbohydrate–electrolyte fluids during physical activity can prevent dehydration and delay the onset of fatigue, allowing for proper biochemical and hematological reactions during exertion. A suitable drinking plan should consider the pre-exercise hydration level as well as the requirements for fluids, electrolytes, and substrates before, during, and after exercise. The objective of this study was to assess the impact of different hydration strategies (isotonic, water, and no hydration) on hematological indicators (hemoglobin concentration, hematocrit number, erythrocyte count, leukocyte count, and mean corpuscular volume) and lactate concentration during prolonged physical exertion in a high-temperature environment in young men. Methods: The research method was quasi-experimental. The study involved 12 healthy men aged 20.6 ± 0.9 years, who were characterized by a body height (BH) of 177.2 ± 4.8 cm, a body mass (BM) of 74.4 ± 7.6 kg, a lean body mass (LBM) of 61.1 ± 6.1 kg, and a body mass index (BMI) of 23.60 ± 0.48. Measurements were taken of body composition and hematological and biochemical indicators. The main tests consisted of three series of tests separated by a one-week break. During the tests, the men performed a 120 min exercise with an intensity of 110 W on a cycle ergometer in a thermo-climatic chamber at an ambient temperature of 31 ± 2 °C. During exertion, the participants consumed isotonic fluids or water in an amount of 120–150% of the lost water every 15 min. The participants who exercised without hydration did not consume any fluids. Results: Significant differences in serum volume were observed between the use of isotonic beverage and no hydration (*p* = 0.002) and between the use of isotonic beverage and water (*p* = 0.046). Immediately after the experimental exercise, hemoglobin values were significantly higher with no hydration than with water (*p* = 0.002). An even stronger significance of differences in hemoglobin was observed between no hydration and isotonic beverage consumption (*p* < 0.001). There was a statistically significant difference in the number of leukocytes between the consumption of isotonic beverage and no hydration (*p* = 0.006). Conclusions: Each active hydration strategy allows for a better maintenance of water–electrolyte homeostasis during physical exertion in a high-temperature environment, and isotonic beverage consumption had a greater impact on hydrating extracellular spaces with the smallest changes in hematological indicators.

## 1. Introduction

Physical work performed in elevated ambient temperature compared to physical effort performed in thermoneutral conditions is a greater challenge for humans and activates individual functional systems to a greater extent [1]. Engaging in physical exertion in an elevated temperature environment leads to water–electrolyte disturbances as well as disruption of thermal balance and internal homeostasis, which enhances the physiological–biochemical changes associated with the physical work [2]. The effectiveness of proper functioning of the body during exertion in elevated temperature depends on a well-functioning circulatory system responsible for transporting heat from the muscles to the skin surface, the amount of sweat produced, body hydration status, blood volume, electrolyte concentrations in bodily fluids, and the level of physical activity performed, which all impact each other [3,4,5,6]. An increased temperature rise during exertion in dehydrated conditions can be accompanied by higher lactate levels in the blood, and may also lead to disturbances in the morphological profile and a decrease in aerobic capacity [3].

Muscular endurance weakness and an increase in internal temperature, as well as an elevated heart rate, further increase the thermal load on the body [7,8]. Therefore, the ability to perform prolonged physical exercise in a warm environment requires replenishment of water losses. Available publications indicate that this effectively improves exercise performance [9,10]. 

Wendt et al. observed positive effects of hydration with carbohydrate–electrolyte fluids during physical exercise [11]. Consumption of carbohydrate–electrolyte beverages can prevent not only dehydration but also enable the proper course of biochemical reactions during exercise [12]. It can be inferred that, in order to maintain the body’s exercise capacity, it is essential not only to hydrate the body but also to provide it with an adequate amount and quality of carbohydrates.

In the case of exercise in elevated ambient temperature, it is important for the body to receive as little heat as possible from the environment. The efficient operation of this mechanism is possible mainly through an increase in ventilation per minute and acceleration of heart rate [6,13]. Heart rate is one of the simplest to monitor and best-studied ways to assess exercise intensity. Almost every study on organism performance involves this indicator, playing an important role in the analysis of the impact of various stimuli on the body. Therefore, the increasing thermal load resulting from dehydration should be reflected in an increase in heart rate [14].

The task of achieving and maintaining an ideal state of hydration during physical activity becomes increasingly challenging, depending on the type of sport, the nature of the activity, and the accessibility of the fluids. While optimal hydration is influenced by various factors, it can generally be described as the prevention of losses exceeding 2–3% of body mass during exercise, while also avoiding excessive fluid intake [15]. In addition, it is not uncommon for individuals to experience unintentional dehydration during exercise due to inadequate fluid consumption relative to their fluid requirements. Conversely, excessive intake of fluids can also present a problem, as severe cases of overhydration can lead to the development of hyponatremia [15]. Inappropriate management of fluid intake resulting in hypohydration, or hyperhydration, can be detrimental for performance and in some circumstances, increases health risk. The reduction in body water levels during physical activity intensifies both physiological and perceptual stress [16]. Additionally, it is widely recognized that such alterations can compromise endurance performance, especially in hot conditions, and may heighten the susceptibility to exertional heat-related illnesses [15].

The decrease in plasma volume during physical work in a high ambient temperature may largely manifest as a decrease in red blood cell volume. Disturbances in the ion balance of the water spaces lead to the penetration of cell water into the extracellular space. The movement of body water from the vascular space to muscle tissue can cause hemoconcentration. It has been shown that an increase in hematocrit (HCT) to 60% can impair blood flow through capillaries, which can have a detrimental effect on the body’s exercise capacity [17,18]. One of the reasons for the limitation of human performance during physical exercise in high ambient temperatures may also be the production of lactic acid in the muscles, which is a by-product of anaerobic metabolic processes. Such processes occur constantly in the body, both during exercise and at rest. Disturbances to the homeostasis of thermal and fluid balance pose a health risk, leading to reduced exercise performance in hot environments. Therefore, athletes are keen to mitigate such performance impairments. In this regard, the aim of this study was to evaluate the effect of different hydration strategies on young men’s hematological indicators, plasma volume, and blood lactate levels during prolonged physical exercise in a high ambient temperature.

## 2. Materials and Methods

The research method was quasi-experimental. A total of 12 healthy individuals were selected as available and purposefully. The study group consisted of 12 selected healthy males with an average age of 20.67 ± 0.98 years, who were characterized by a body height (BH) of 177.25 ± 4.83 cm, a body mass (BM) of 74.45 ± 7.6 kg, a lean body mass (LBM) of 61.18 ± 6.19 kg, characterized by an average level of aerobic fitness according to American Heart Association norms (2003). The male subjects were informed about the experiment conditions and gave written consent to participate in the study. The research project obtained approval from the Bioethics Commission at the District Medical Chamber in Krakow (No. 42/KBL/OIL/2015). With the approval of the unit supervisor, the study was conducted at the Laboratory of Physiological Adaptation, Academy of Physical Education in Krakow (PN-EN ISO 9001: Kobierzyce, Poland, 2015According to the requirements of the Helsinki Declaration, the men were informed about the purpose of the study, the used methodology’s possible side effects, and the option to withdraw from the study at any stage without giving a reason. The entire experiment was carried out under the supervision of qualified medical personnel in accordance with applicable standards. During the period of the experiment, study participants did not use any stimulants, vitamins, or other supplements. A total of 12 people completed the full cycle of tests.

### 2.1. Research Methods

The study was divided into two main stages: preliminary and main research (Figure 1).

The first stage of the project involved medical examinations and measurements of morphological indicators of body composition, including body height (BH) measured with a Martin anthropometer (Seritex, Ney York, NY, USA) with an accuracy of 0.1 cm and body mass (BM) measured with a Sartorius F 1505-DZA electronic scale (in Göttingen, Germany) with an accuracy of 1 g. Based on the obtained data, the Quetelet index was calculated. Body composition was also estimated using an eight-electrode JAWON MEDICAL IOI-353 body composition analyzer (EC0197-Seoul, South Korea) with the bioelectrical impedance analysis method, including the measurement of percentage body fat (PBF), fat mass (FM), and fat-free mass (LBM).

### 2.2. Wingate Test

The Wingate test was used to assess anaerobic performance. Prior to the main effort, a 5 min warm-up was performed on a cycle ergometer with individually selected intensity, performed at a rhythm of 60 rev/min with a load of 40% of maximal oxygen uptake, determined earlier during a graded exercise test with three 5 s accelerations of pedaling rate to a speed close to maximum at 2, 4, and 5 min. Two minutes after the warm-up, the men performed a 30 s maximal physical effort with a load of 8.3% of body mass [19], during which the following indicators were recorded: peak power (PP), mean power (MP), total work (TW), time to reach peak power (toPP), time to maintain maximal power (tmPP), and the power drop index (IDP). The anaerobic test (Wingate test) was performed on a Monark 875E cycle ergometer.

#### 2.2.1. Graded Exercise Test on a Cycle Ergometer

A graded exercise test “to subjective refusal” was performed at an ambient temperature of 21 ± 0.5 °C and relative humidity of 40 ± 3%. The purpose of the test was to determine individual ventilatory thresholds, which were used to determine the appropriate loads for the main research tests. The exercise was performed on a cycle ergometer. A three-minute warm-up with a pedal frequency (RPM) of 60 revolutions per minute and intensity of 110 W was conducted, followed by an increase in power by 30 W every 2 min. The exercise continued until the subjectively perceived inability to maintain the desired pedaling rhythm, or the refusal to continue the exercise at the specified pedaling resistance. During the exercise, indicators of respiratory exchange, such as oxygen uptake per minute ($\dot{V}$O_2_), minute carbon dioxide output ($\dot{V}$CO_2_), ventilation per minute ($\dot{V}$E), and respiratory exchange ratio (RER), were recorded every 30 s using a spirometer. Heart rate (HR) was continuously recorded.

Respiratory physiological indicators in the graded exercise test were recorded in thirty-second sequences using a computerized Ergospirometry START 2000M apparatus from MES (Poland). This model is equipped with analyzers of oxygen (O_2_) and carbon dioxide (CO_2_). Based on these measurements, variables including oxygen uptake per minute ($\dot{V}$O2), minute CO_2_ output ($\dot{V}$CO_2_), fractional concentration of oxygen in expired air (FeO_2_), fractional concentration of carbon dioxide in expired air (FeCO_2_), ventilation per minute ($\dot{V}$E), respiratory exchange ratio (RER), carbon dioxide equivalent for ventilation ($\dot{V}$E/$\dot{V}$CO_2_^−1^), oxygen equivalent for ventilation ($\dot{V}$E/$\dot{V}$O_2_^−1^), respiratory rate (fR), and tidal volume (VT) were determined. The graded exercise test was performed on an ER 900 D-72475 BIT2 cycle ergometer from Jaeger (Duisburg, Germany). Continuous exercise performed in a thermoclimatic chamber (120 min, at an ambient temperature of 31 ± 20 °C, relative humidity of 60 ± 3%, and air velocity below 0.5 m·s^−1^, with physical load constant and set at 53 ± 1% $\dot{V}$O_2_peak), which allowed for the determination of post-exercise water loss. The degree of dehydration was determined based on body mass measurements (with an accuracy of 1 g) before and after the exercise tests, as well as taking into account the potential volume of urine excreted (Sartorius F 150S—DZA balance, in Göttingen, Germany).

The second stage of the experiment consisted of three series of weekly tests, separated by a one-week break. During each series, the participants performed a 120 min constant load exercise at 53 ± 1% $\dot{V}$O_2_peak in an ambient temperature of 31 ± 20 °C, relative humidity of 60 ± 3%, and air movement below 1 (m s^–1^). The test began with a five-minute acclimatization to the environmental conditions in the chamber. After this stage, the participants immediately began the actual exercise test. The ambient temperature and relative humidity in the thermoclimatic chamber were controlled using a Harvia (Muurame, Finland) thermo-hygrometer and an Ellab (Hillerød, Denmark) electro-thermometer with an accuracy of ± 3% and ± 0.5 °C, respectively, while air movement was measured using a Hilla catheter thermometer.

The physical load was constant and set at 53 ± 1% $\dot{V}$O_2_peak. Each experimental test series was separated by a one-week break necessary to extinguish any potential effects of physical exertion. Additionally, a crossover design experiment [16] was applied during the tests, in which each possible order of using different hydration strategies appeared an equal number of times (Table 1). This eliminated the potential impact of using different hydration strategies in a particular order on the results of the experiment.

Heart rate (HR) during laboratory tests was recorded telemetrically using a Polar 610S heart rate monitor from Polar Electro (Finland). Blood samples for hematological and biochemical measurements were taken directly before the start of the 120 min exercise trial and during selected phases of recovery, i.e., between 3 and 5 min and between 55 and 60 min, as well as at 24 and 48 h after exercise. Venous blood samples were collected for hematological measurements. Blood samples for lactate concentration measurements were also taken directly before, as well as at 3 and 20 min after the exercise trial. Blood samples for biochemical and hematological measurements were collected in accordance with current standards by a certified laboratory diagnostician. Each pair in the crossover design completed consecutive weeks of testing with a different hydration strategy (Figure 2).

During exercise, the participants (strategy—I) consumed isotonic fluids (with a composition of osmolality—270–330 mOsm/kg of water, carbohydrate content of 6–8 g per 100 mL of beverage, and sodium (Na^+^) content of 20–50 mmol/L (i.e., 460–1150 mg/L)), or water (strategy—W) in an amount of 120–150% ΔBM [20,21]. The hydration strategies were based on standards adopted by the American College of Sports Medicine (ACSM). According to ACSM (Indianapolis, IN, USA 1996), a person engaging in physical activity should consume beverages at a temperature of 13–15 °C, every 15–20 min, in a volume of 150–300 mL. Participants who engaged in physical activity without hydration (strategy—NH) did not consume any fluids (Table 1 and Figure 2).

#### 2.2.2. During the Recovery Period

After the 120 min exercise, the participants remained at room temperature for 90 min. During rest, the participants (strategy—I) consumed isotonic fluids, water (strategy—W), or no fluids (strategy—BN). The volume of fluids to be consumed after exercise was determined based on the individual water loss recorded during preliminary testing (Figure 1).

Additionally, before exercise (120 and 10 min before the trial), the participants consumed isotonic beverages or water (strategies—I and W) according to ACSM standards, (Indianapolis, IN, USA 1996).

### 2.3. Assessment of Hematological and Biochemical Indices

Blood samples for hematological and biochemical measurements were collected directly before the test (preT), immediately after the test (0), and at 1, 24, and 48 h after exercise. For hematological measurements, 4 mL of blood was collected from the antecubital vein using K_2_EDTA-containing tubes. The collected material was used to determine the following morphological indices: hemoglobin concentration (HB), hematocrit number (HCT), erythrocyte count (RBC), leukocyte count (WBC), and mean corpuscular volume (MCV). The above hematological and biochemical measurements were outsourced as external tests. Immediately after the material was collected, it was transported at the appropriate temperature to the Diagnostic Laboratory Center “DIAGNOSTYKA” branch in Krakow, ul. Prof. M. Życzkowskiego 16.

Blood samples for subsequent biochemical measurements were also collected from the antecubital vein into 6 mL tubes containing a clot activator. To obtain the necessary material in the form of serum, the collected blood sample had to be centrifuged using a laboratory centrifuge (Medical Instruments Warsaw, Poland, Centrifuge MPW 351R). The obtained serum was then transferred to Eppendorf-type tubes. The prepared material was frozen at −20 °C (Zanussi KBI300, Warsaw, Poland). After 24 h, the material was transferred to a freezer maintaining a cooling temperature of −70 °C (Artico ULF 390 ChRL freezer, Esbjerg, Denmark).

Measurements that were not outsourced were performed at the Academy of Physical Education in Krakow, in the Department of Biomedical Sciences, Physiology and Biochemistry Laboratory, using the E-liza Mat 3000 microplate reader (DRG, Medical Instruments GmbH, Marburg, Germany).

Arterialized blood was used to determine lactate concentration, which was collected from the fingertip using a heparinized capillary with a volume of 10 μL. Collection was performed directly before exercise and 3 and 20 min after its completion. The determination was carried out using a minifotometer plus Dr Lange, type LP-20 from Dr Lange (Berlin, Germany).

Plasma volume changes (%ΔPV) were estimated using the Dill and Costill formula (1974) with modifications by Harrison et al. [22]. The calculations were made based on the results of Hb and Hct using the following formula:%ΔPV = 100{(Hb1/Hb2)·[100 − (Hct2·0.874)]/[100 − (Hct1·0.874)]^−1^}
(where Hb1 and Hct1 are the initial values of hemoglobin concentration and hematocrit number, and Hb2 and Hct2 are the values of these indices after exercise). Post-exercise concentration values were corrected for changes in plasma volume.

### 2.4. Statistical Analysis

Descriptive statistics were used to analyze the results of the measured morphological structure, as well as aerobic and anaerobic capacity of the study group, with the results presented as means ± standard deviation (x ± SD). The impact of watering strategy, measurement order, and their interaction (HS, Trial, HSxTrial) on the mean values of the tested indicators were examined using a repeated measures analysis of covariance (ANCOVA). The SAS 9.3 software was utilized for the analysis, employing a mixed model (Generalized Linear Mixed Model—SAS MIXED procedure, Cary, North Carolina, USA.) with the REML estimation method. The analysis included two models concurrently: one assuming equal variances for three consecutive tests (compound symmetry), and a more intricate model (unstructured) that often produces more robust results. The LR test was applied to compare the fitness of the models with differing variance structures, with the LR test outcomes providing a definitive response concerning the validity of utilizing a simpler model with fewer parameters, namely a model assuming compound symmetry. The mean values were estimated utilizing the least squares approach and adjusted using the Tukey–Kramer post hoc test, with the analysis performed both with and without remaining data for the subject in cases of exclusion due to errors during sample collection, incorrect labeling, or outliers. The fixed effects were considered statistically significant when *p* < 0.05, with the results presented in the form of tables containing the analysis of variance for each variable and estimated *p*-values for each comparison, adjusted using the Tukey–Kramer post hoc test. The interaction of fixed effects was considered significant only when the *p*-value was <0.01, due to the large number of variables and relatively small sample size.

## 3. Results

The male participants, aged 20.67 ± 0.98 years, were characterized by a height (BH) of 177.25 ± 4.83 cm, a body mass (BM) of 74.45 ± 7.6 kg, and a lean body mass (LBM) of 61.18 ± 6.19 kg. The aerobic capacity ($\dot{V}$O_2_peak) of the male participants was 3.70 ± 0.70 L·min^−1^, and in relative terms, 49.70 ± 6.70 mL·min^−1^·kg^−1^. The anaerobic capacity (RPP) was 827.17 ± 97.99 W and 11.07 ± 0.78 in relative values (Table 2).

Regardless of the hydration strategy used, the mean body mass (BM) of the subjects decreased during exercise. The greatest decrease in body mass was observed when water was used as the hydration strategy, but the differences between the groups were not statistically significant (Table 3 and Figure 3).

The change in plasma volume (ΔPV) measured directly after and one hour after the exercise test was significantly dependent on the hydration strategy (*p* = 0.003 and *p* = 0.002, respectively). Immediately after exercise, the isotonic drink did not result in a decrease in PV, and even a slight increase was observed. A significant decrease in plasma volume was observed with water hydration, and the greatest decrease was observed with no hydration. Significant differences in plasma volume were observed between isotonic drink and no hydration (*p* = 0.002). An hour after exertion, a significant increase in plasma volume was observed when an isotonic drink was consumed. An increase in PV was also observed when water was consumed. Lack of hydration maintained a negative balance of plasma volume (Table 3 and Figure 4).

The mean concentration of hemoglobin (HB) differed significantly depending on the hydration strategy used immediately after and one hour after the exertion test (Table 3). Immediately after the exertion test, the HB values were significantly higher when no hydration was used compared to when water was used (*p* = 0.002). One hour after the test, a significantly higher HB value was recorded when no hydration was used compared to when water or isotonic drink was used (*p* < 0.001) (Table 3 and Figure 5).

The mean values of hematocrit (HCT) measured one hour after the exertion test were significantly higher when no hydration was used compared to when water was used (*p* = 0.003), as well as significantly higher when no hydration was used compared to when isotonic drink was used (*p* = 0.002) (Table 3 and Figure 5).

The volume of erythrocytes (MCV) did not differ depending on the hydration strategy in any of the four measurements. One hour after exercise, a slight decrease in MCV can be observed with the lack of hydration and use of water (Table 3 and Figure 6).

The mean number of red blood cells (RBCs) measured before the test did not differ significantly depending on the hydration strategy used. Immediately after the test, RBC values were highest with no hydration, moderate with isotonic drink, and lowest with water. Significant differences in RBC were observed between no hydration and isotonic drink (*p* = 0.027), as well as no hydration and water (*p* = 0.016). One hour after exercise, the RBC levels were significantly higher with no hydration than with water or isotonic drink (*p* < 0.001). In the measurement taken 24 h after exercise, there were no statistically significant differences in mean RBC values depending on the hydration strategy used (Table 3).

The mean white blood cell count (WBC)/Leukocytes measured before the test did not differ significantly depending on the hydration strategy used. However, the hydration strategy had a significant effect on the WBC count measured immediately after, as well as 1 h and 24 h after exercise. Immediately after and one hour after exercise, the WBC count was highest when water was used, and lowest when an isotonic drink was used. The analysis of results from the last measurement is complicated by a significant interaction between the effects of hydration and the order of performing the test (Table 3 and Figure 7).

The lactate concentration (LA) in each of the measurements was slightly higher when using water compared to no hydration, and lowest when using isotonic drink. However, these differences were not statistically significant, both in measurements performed before and after the test (Table 3 and Figure 8).

The results for measurements taken at the 80th minute of exercise are complicated by a significant interaction of the effects of hydration strategy and test order (Table 4 and Figure 9). In the 80th minute of the first test (T1), HR was significantly higher with no hydration compared to both isotonic drink (*p* = 0.024) and water (*p* = 0.021) hydration. In the second test (T2), the highest HR was observed with isotonic drink and was significantly higher than with water (*p* = 0.012), but not different from no hydration (*p* = 0.714). Significantly higher HR was observed with no hydration compared to water (*p* = 0.039). In the third test (T3), HR values were similar and not significantly different (*p* = 0.557; Table 4) (Figure 9).

The average HR values measured in the 40th and 115th minute of the test were highest with no hydration. The lowest HR values were observed with water hydration. However, the only statistically significant difference was between water and no hydration in the 115th minute (*p* < 0.001).

HR measured in the 80th minute of T1 was significantly higher with no hydration than with isotonic drink (*p* = 0.024) and equally significantly higher than with water (*p* = 0.021) (Table 4). In T2, the highest value was observed with isotonic drink and was significantly higher than with water (*p* = 0.012). Significantly higher HR was observed with no hydration compared to water (*p* = 0.039). In the last test (T3), values were similar and not significantly different. The highest HR was observed with water, slightly lower with no hydration, and lowest with isotonic drink (Table 4).

Higher values of WBC concentration were recorded when using water compared to no hydration or isotonic drink immediately and one hour after exercise. Immediately after exercise, significantly higher WBC values were observed when using water compared to isotonic drink (*p* = 0.002). The difference in WBC values was also statistically significant one hour after exercise (*p* = 0.002) (Table 5). In the 24 h period after T1, noticeably higher WBC values were recorded when using isotonic drink compared to water or no hydration. The difference in WBC values was statistically significant between using isotonic drink and no hydration (*p* = 0.006). In T2, the highest WBC values were recorded with no hydration, which were significantly higher than values recorded with isotonic drink (*p* = 0.006) or water (*p* < 0.001). In T3, the highest WBC values were observed with isotonic drink, medium with water, and the lowest with no hydration. However, these differences were not statistically significant (Table 5).

## 4. Discussion

The aim of this study was to investigate the effectiveness of different hydration strategies (carbohydrate–electrolyte beverage, water, or no hydration) during exercise in high ambient temperatures on hematological indices and lactate concentration in the body. Based on the results of the study, it was shown that hydration with a carbohydrate–electrolyte beverage compared to water was more effective in regulating the permeation of body water between intra- and extracellular spaces, resulting in a smaller loss of serum volume. Significant differences in serum volume were observed between the use of isotonic beverage and no hydration and between the use of isotonic beverage and water. Immediately after the experimental exercise, hemoglobin values were significantly higher with no hydration than with water. An even stronger significance of differences in hemoglobin was observed between no hydration and isotonic beverage consumption. There was a statistically significant difference in the number of leukocytes between the consumption of isotonic beverage and no hydration. Both the domestic and foreign literature provides a considerable amount of scientific reports on the negative impact of high ambient temperature on the physical performance of individuals engaging in physical activity under such conditions [23,24]. After analyzing the studies conducted so far, it can be inferred that the main cause of the loss of the body’s work efficiency in high ambient temperatures is dehydration associated mainly with sweating. Many authors of publications prove that dehydration equivalent to a loss of approximately 2% of body mass causes significant water–electrolyte disturbances, which can impair physiological and biochemical processes in the human body and, as a result, reduce exercise performance [6,25]. The aim of this study was to investigate the effectiveness of different hydration strategies (carbohydrate–electrolyte beverage, water, or no hydration) during exercise in high ambient temperatures on hematological indices and lactate concentration in the body.

Rodriguez et al. [26] investigated the effect of isotonic drinks (consumed before and during exercise) on the cardiovascular responses of trained and untrained men during three different workloads in a hot environment. They demonstrated that the consumption of isotonic drinks, as opposed to no hydration, significantly reduced exercise heart rate (HR) in both the 40% and 60% $\dot{V}$O_2_peak workload tests. These results are consistent with the present study. Both water and isotonic drink hydration strategies resulted in lower HR values at the 40 and 115 min time points compared to no hydration, with a statistically significant difference in HR observed between water and no hydration at 115 min into the exercise. This suggests that the body may not have been able to cope with increasing thermal stress and dehydration at this stage of the exercise. It should be noted that the mean HR did not significantly differ throughout the entire exercise period between the water and isotonic drink groups. Based on our own study, it can be assumed that the reduction in exercise HR was not dependent on the composition of the hydration drink, although blood glucose levels were not analyzed. The lower HR level was most likely due to replenishing fluid deficits.

The results of this study show that drinking water before, during, and after exercise is more or equally effective in reducing heart rate compared to isotonic drinks. Prolonged exposure to endogenous and exogenous heat stress leads to increasing dehydration of the body, which was observed in the studied men regardless of the hydration strategy used. This was mainly reflected in the decrease in body mass, which averaged 0.8 kg. The largest decrease was observed when water was used—on average, 0.94 kg. Previous studies have shown that exercise performed under such conditions causes fluid loss primarily from the extracellular space, followed by the intracellular space [1,27,28]. In our own research, it was observed that intense and prolonged sweating can lead not only to a decrease in PV levels, causing an increase in HCT and RBC, but also to an increase in the number of leukocytes. Similar effects were observed in Blum’s study [29].

Petersen et al. [28] showed that heat stress causes similar changes in the white blood cell profile as physical exertion. In our own research, we observed an increase in leukocyte concentration immediately after and an hour after exercise for all three hydration strategies. After 24 h, their values practically returned to pre-test levels. It can be concluded that experimental stress caused leukocytosis, which lasted no longer than 24 h. It should be emphasized that significantly smaller increases in leukocyte concentrations were noted when using isotonic drinks compared to water or no hydration.

The change in body mass recorded in the experiment, caused by exercise in elevated temperature, was associated with loss of body water due to increased sweat production. As a result, changes in the effective molality of body fluids and increased water penetration from the intracellular to the extracellular space may have occurred. This is a widely recognized cause of decreased exercise performance [30]. However, the use of isotonic drinks appears to effectively inhibit extracellular water loss. The results of our own research showed that, immediately after exercise, while using isotonic drinks, there was no decrease in plasma volume, and even a slight increase was noted. A significant decrease in PV was observed when using water, about 2%, and the largest decrease of up to 4% was noted with no hydration. The differences between the use of isotonic drinks and water or no hydration were statistically significant. An hour after exercise, a clear increase in plasma volume was observed when using isotonic drinks. The use of water also resulted in an increase in PV, although in this case, despite using the same hydration protocol during recovery, it was not as effective as isotonic drinks. Lack of hydration maintained a negative balance of plasma volume. A decrease in plasma volume can largely manifest itself in a decrease in red blood cell volume. As a result of disturbances in the ionic balance of water spaces, intracellular water penetrates into extracellular spaces. Previous studies have shown that aerobic exercise, during which osmolality increased slightly and pH change was also small (<0.1), led to erythrocyte contraction due to dehydration [31].

My own research shows that, regardless of the hydration strategy used, experimental exercise resulted in a decrease in erythrocyte volume, although the changes were small. Significant differences in the effectiveness of different hydration strategies could be observed immediately after exercise. The smallest changes were observed when using an isotonic drink, moderate changes when using water, and the largest changes when no hydration was provided. These relationships were also observed one hour after exercise, but the average ΔMCV differences were small and not statistically significant. It is worth noting that even 24 h after the exercise trial, erythrocyte volumes did not return to pre-exercise values regardless of the hydration strategy used. In men who did not receive hydration during the test, a clearly higher degree of erythrocyte dehydration was observed at 24 h compared to when an isotonic drink or water was used, even though other parameters such as plasma volume or body mass returned to pre-test values or were even higher. This research shows that, regardless of the hydration strategy used, experimental exercise caused a decrease in erythrocyte volume, although the changes were small. Significant differences in the effectiveness of different hydration strategies could be observed immediately after exercise. The smallest changes were noted when using an isotonic drink, medium when administering water, and the largest when no hydration was used. Such dependencies were also noted an hour after exercise, but the average ΔMCV differences were small and not statistically significant. It is worth noting that, even 24 h after the exercise trial, erythrocyte volumes did not return to pre-exercise values regardless of the hydration strategy used. In men who did not hydrate during the test, a clearly higher degree of dehydration of red blood cells was observed at 24 h than when using an isotonic drink or water, even though other parameters such as plasma volume or body mass returned to pre-test values or even increased.

The use of an isotonic drink probably maintains the proper concentration of osmotically active compounds, and such a state inhibits the transport of water resources from the intracellular space to the outside of the cells, thus preventing their dehydration. It can therefore be assumed that isotonic drinks also effectively eliminate heat from the body, but at the same time do not cause extracellular and intracellular dehydration, as evidenced by the results of changes in erythrocyte volume.

It has been shown that increasing hematocrit (HCT) to 60% can impair flow through capillaries, which can adversely affect the body’s exercise capacity. The authors of [18,32] also found a significant increase in HCT and protein concentration as a result of intense exercise by athletes (70% $\dot{V}$O_2_peak, 180 min). Analysis of the results of the study showed that the exercise did not raise hematocrit above 45% in any of the tested groups, so blood density should not significantly impede its flow through capillaries. Immediately after exercise, the lowest HCT level was observed when using an isotonic drink, medium when using water, and the highest when no hydration was used. The largest differences between hydration strategies were observed an hour after exercise. The HCT level when using water and an isotonic drink was almost equal, while with no hydration, it was significantly higher, which could affect the transport efficiency of blood and contribute to less effective regeneration. It is worth noting that the level of HCT changes (although not statistically significant) showed that the lack of hydration caused the greatest increase in this indicator, and especially an hour after exercise, the use of an isotonic drink clearly lowered the hematocrit count. Physical activity has been observed to generally correlate with elevated HDL cholesterol and reduced levels of LDL cholesterol and triglycerides. Exercise not only induces quantitative modifications in serum lipids, but also has a favorable effect on the maturation, composition, and functionality of HDL particles [33,34].

Similar conclusions can be drawn when analyzing the results of changes in hemoglobin concentration (ΔHB) and red blood cells (ΔRBC). The isotonic drink proved to be the most effective in maintaining or lowering the levels of these indicators immediately after exercise or after one hour of rest and rehydration. Water use was less effective, although an hour after exercise it caused a decrease in concentrations below the pre-exercise level (preT). The lack of hydration had the least favorable effect on the height of changes in RBC and HB concentrations. Immediately after exercise, it showed the greatest increase, and after one hour of recovery, it did not cause the concentrations to return to preT values. With the no hydration strategy, a decrease in concentrations 24 h after exercise was observed compared to preT values, which may indicate increased rehydration in the subjects who took fluids in any way throughout the day after the test and planned rest (90 min).

The obtained results confirm that, when an isotonic drink is consumed, there are no significant changes in osmolal pressure between the body’s water spaces. The volume of plasma did not decrease significantly, and the oncotonic pressure of plasma protein colloid did not increase, thus reducing the dehydration of red blood cells. Therefore, the use of an isotonic drink is more effective than water in maintaining exercise performance, positively affecting blood fluidity and cell elasticity, especially erythrocytes. Analysis of lactate concentrations in the tested men showed that, even before the exercise test, its levels fluctuated between 1.9 and 2.2 mmol·L^−1^, a range of values commonly recognized as resting. After exercise, the concentration of lactate in venous blood increased to a maximum of 3.0 mmol·L^−1^, indicating the aerobic nature of the experimental exercise. Changes in lactate concentrations were small and did not differ significantly depending on the hydration strategy used, although it is worth noting that, when using an isotonic drink, they were noticeably smaller than when using water or no hydration. It can be concluded that the load applied at 53% $\dot{V}$O_2_peak fulfilled the expected role, and such a slight increase in lactate concentration during exercise could not cause a defensive reaction of the body and stop the resynthesis of adenosine triphosphate (ATP) [35].

## 5. Conclusions

The use of an isotonic drink compared to drinking water or no hydration resulted in a smaller decrease in plasma volume and less dehydration of erythrocytes, indicating its effectiveness in maintaining osmotic balance and inhibiting the movement of body water between compartments.

## 6. Practical Implication

The isotonic drink was the most effective way to hydrate the extracellular space, causing the least changes in hematological indicators.

## Figures and Tables

**Figure 1 biology-12-00687-f001:**
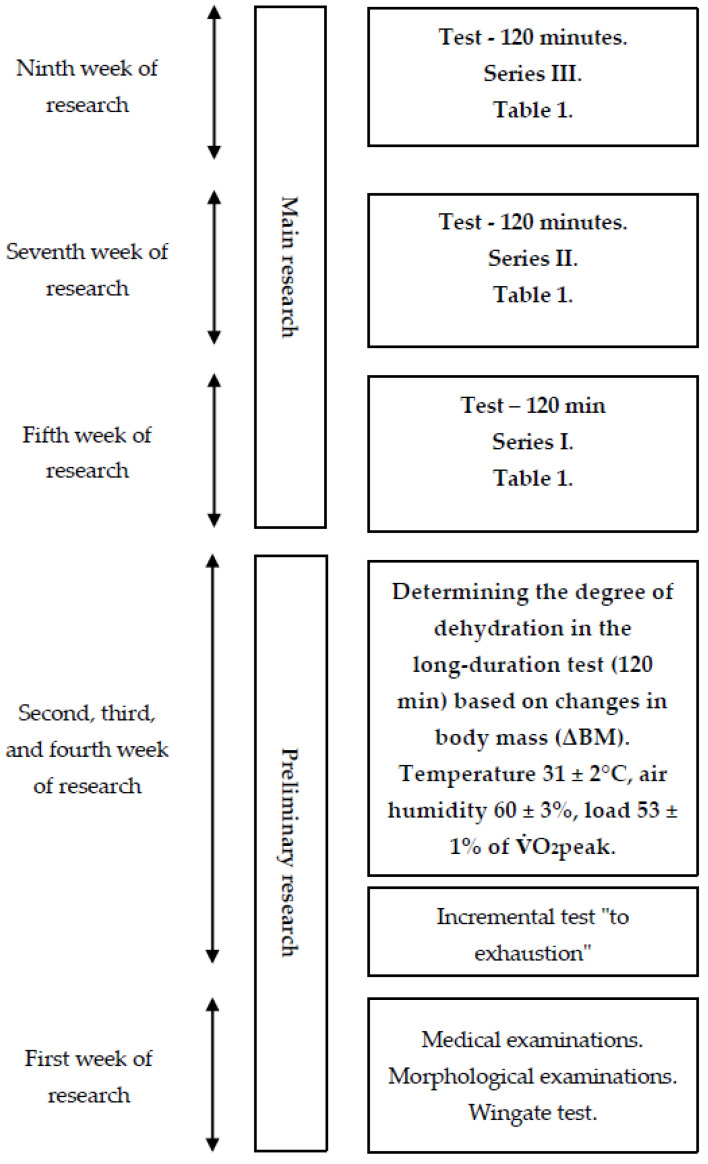
General research scheme.

**Figure 2 biology-12-00687-f002:**
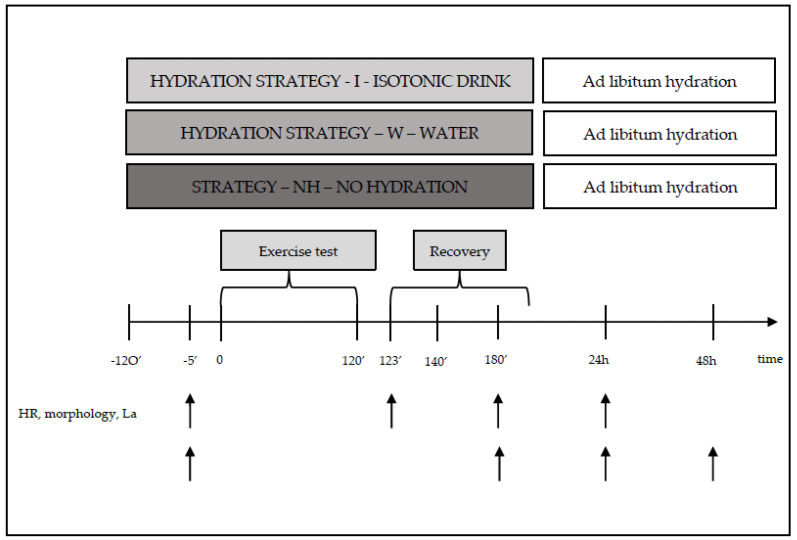
Diagram of one series of tests during the main study, along with measurement points.

**Figure 3 biology-12-00687-f003:**
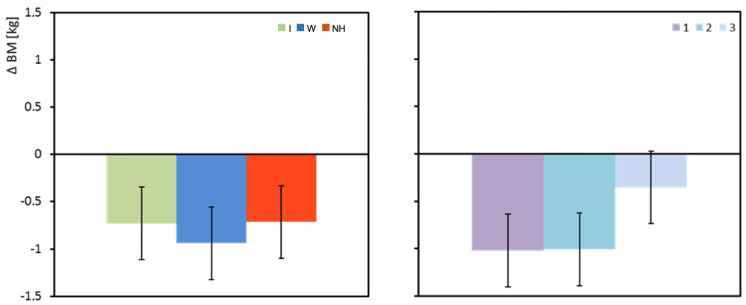
Change in body mass between pre- and post-test (ΔBM). Mean values estimated by the least squares method with upper and lower confidence intervals. Measurements were performed on 12 men in three consecutive tests (1, 2, and 3) using three hydration strategies (I—isotonic drink, W—water, NH—no hydration).

**Figure 4 biology-12-00687-f004:**
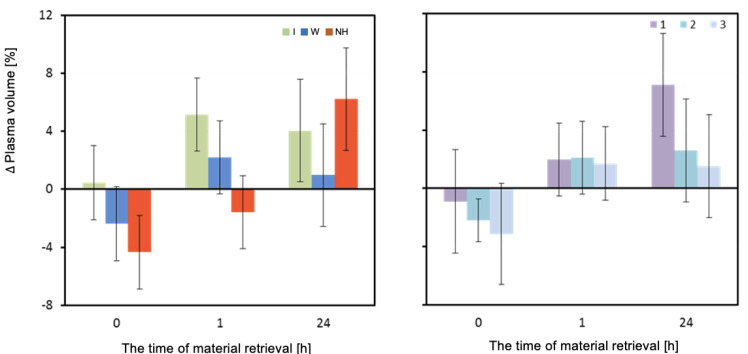
Change in plasma volume ΔPV (%). Mean values estimated by the least squares method with upper and lower confidence intervals for three measurements (between preT value and 0, 1, and 24 h after the test) performed on 12 men in three consecutive tests (1, 2, 3) using three hydration strategies (I—isotonic drink, W—water, NH—no hydration).

**Figure 5 biology-12-00687-f005:**
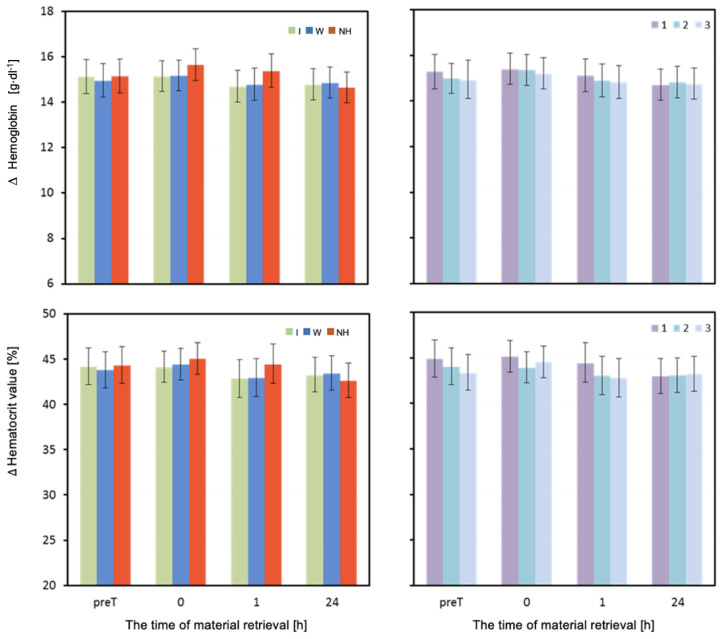
Hemoglobin and hematocrit. Mean values estimated by the least squares method with upper and lower confidence intervals for four measurements (immediately before the test—preT, after the test—0, 1, and 24 h) performed on 12 men in three consecutive tests (1, 2, 3) using three hydration strategies (I—isotonic drink, W—water, NH—no hydration).

**Figure 6 biology-12-00687-f006:**
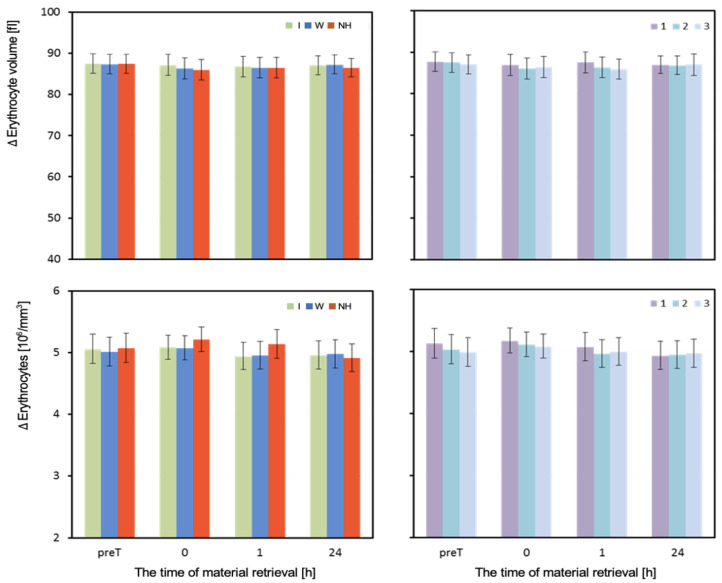
Erythrocyte volume and concentration. Mean values estimated by the least squares method with upper and lower confidence intervals for four measurements (before the test—preT, and at 0, 1, and 24 h after the test) performed on 12 men in three consecutive tests (1, 2, 3) using three hydration strategies (I—isotonic drink, W—water, NH—no hydration). Values restored to the original scale after prior logarithmic transformation.

**Figure 7 biology-12-00687-f007:**
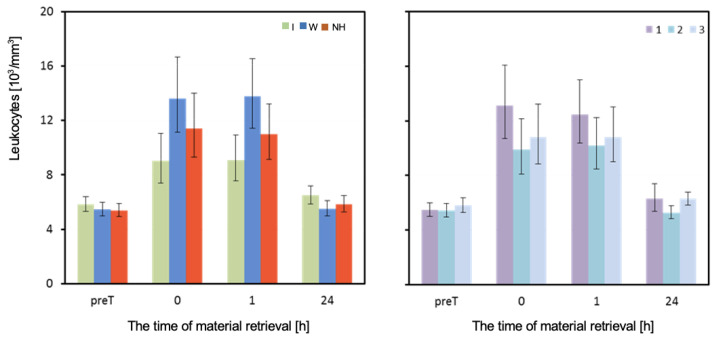
Leukocytes. Mean values estimated by the least squares method along with upper and lower confidence intervals for four measurements (immediately before the test—preT, after the test—0, 1, and 24h) carried out on 12 men in three consecutive tests (1, 2, 3) using three hydration strategies (I—isotonic drink, W—water, NH—no hydration). Values restored to the original scale after previous logarithmic transformation.

**Figure 8 biology-12-00687-f008:**
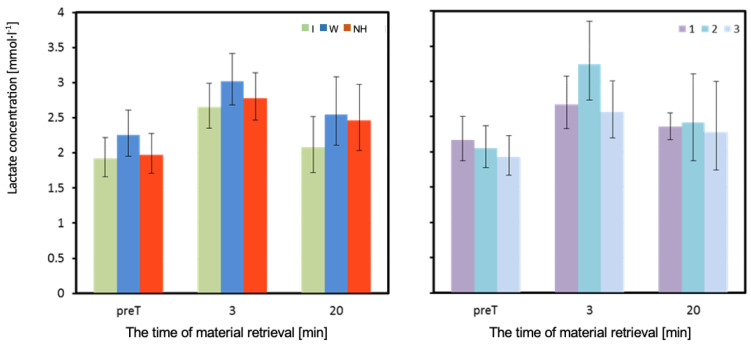
Concentration of lactate in venous blood. Mean values estimated by the least squares method with upper and lower confidence intervals for three measurements (immediately before the test—preT, 3, and 20 min after the test) performed on 12 men in three consecutive tests (1, 2, 3) using three hydration strategies (I—isotonic drink, W—water, NH—no hydration). Values restored to the original scale after previous logarithmic transformation.

**Figure 9 biology-12-00687-f009:**
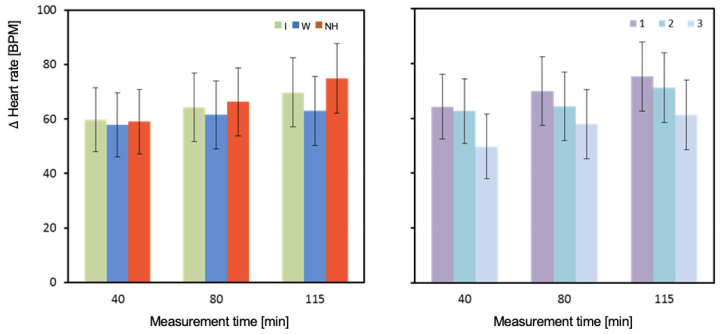
Change in heart rate between pre- and post-test (ΔHR_avg3). Mean values estimated by the least squares method with upper and lower confidence intervals. Measurements were performed on 12 men in three consecutive tests (1, 2, and 3) using three hydration strategies (I—isotonic drink, W—water, NH—no hydration).

**Table 1 biology-12-00687-t001:** Experimental design using a crossover design, in which six pairs of participants performed tests in three hydration conditions (I—isotonic drink, W—water, NH—no hydration) with one-week intervals between each series, with an equal number of treatments in each of the six possible sequences.

Group	Number of Participants	Hydration in Subsequent Tests		
		Test I	Test II	Test III
I-W-NH	2	I	W	NH
W-NH-I	2	W	NH	I
NH-I-W	2	NH	I	W
I-NH-W	2	I	NH	W
W-I-NH	2	W	I	NH
NH-W-I	2	NH	W	I

**Table 2 biology-12-00687-t002:** Descriptive statistics for somatic parameters, the stepwise refusal test, and the Wingate test.

Variable	x	SD
AGE (years)	20.7	1.0
BODY HEIGHT (cm)	177.3	4.8
BODY MASS (kg)	74.5	7.6
LEAN BODY MASS (kg)	61.2	6.2
BODY FAT MASS (kg)	13.3	2.4
BODY FAT PERCENTAGE (%)	17.8	2.2
TOTAL BODY WATER (kg)	44.1	4.4
BODY MASS INDEX (kg·m^2^)	23.7	2.1
tmax (min)	14.6	2.6
PP (W)	291.1	42.1
HRmax (ud·min^−1^)	187.8	6.7
$\dot{V}$O_2peak_ (L·min^−1^)	3.7	0.7
$\dot{V}$O_2peak_ (mL·min^−1^·kg^−1^)	49.7	6.7
$\dot{V}$Emax (L·min^−1^)	132.8	43.6
tVT2 (min)	7.6	2.0
HRVT2 (ud·min^−1^)	147.0	15.0
$\dot{V}$EVT2 (L·min^−1^)	62.8	17.8
PVT2 (W)	185.6	33.9
$\dot{V}$O_2_VT2 (L·min^−1^)	2.4	0.5
$\dot{V}$O_2_VT2 (mL·min^−1^·kg^−1^)	32.7	6.1
%HRmaxVT2	78.2	6.7
% $\dot{V}$O_2_peakVT2	65.9	6.7
BM (kg)	74.8	7.5
Ttest (s)	30.0	0.0
Load (kg)	6.2	0.6
MP (W)	663.1	72.0
MP (W·kg^−1^)	8.9	0.5
TW (kJ)	19.9	2.2
TW (J·kg^−1^)	266.3	16.1
PP (W)	827.2	98.0
RPP (W·kg^−1^)	11.1	0.8
IDP (W·kg^−1^·s^−1^)	0.2	0.0
toPP (s)	4.3	0.5
tmPP (s)	4.2	1.4

**Table 3 biology-12-00687-t003:** Results of analysis of variance.

Variables			Hydration Strategy			Test Order			Interaction		
				(DFl = 2)			(DFl = 2)			(DFl = 4)	
			DFm	F	*p*	DFm	F	*p*	DFm	F	*p*
Body Mass (ΔBM)			17.7	0.39	0.683	17.7	3.59	0.049	17.2	1.46	0.257
Heart Rate	Time	preT	16.5	0.36	0.703	16.5	1.10	0.355	20.0	1.42	0.263
		40	16.2	0.54	0.595	16.2	11.47	**0.001**	20.3	1.55	0.225
		80	11.8	6.76	**0.011**	7.7	56.62	**<0.001**	8.4	10.03	**0.003**
		115	11.6	14.97	**0.001**	6.8	17.70	**0.002**	6.6	4.69	**0.041**
Plasma Volume (ΔPV)	Time	0	9.4	11.65	**0.003**	8.0	2.14	0.181	11.3	3.80	**0.035**
		1	15.5	9.54	**0.002**	15.5	0.03	0.968	27.0	1.07	0.390
		24	15.3	2.41	0.123	15.3	3.01	0.079	23.7	0.52	0.723
Hemoglobin Concentrations	Time	preT	11.1	2.39	0.137	9.3	4.99	**0.034**	9.4	2.15	0.153
		0	16.0	11.24	**0.001**	16.0	1.33	0.291	16.9	3.48	**0.030**
		1	16.0	17.07	**0.000**	16.0	2.91	0.084	16.9	2.57	0.076
		24	16.2	0.86	0.444	16.2	0.25	0.780	17.6	0.16	0.957
Hematocrit	Time	preT	16.1	0.91	0.421	16.1	7.60	**0.005**	17.1	0.24	0.913
		0	15.9	1.42	0.270	15.9	2.38	0.124	19.0	1.58	0.221
		1	15.9	11.03	**0.001**	15.9	10.75	**0.001**	16.8	1.51	0.243
		24	16.1	1.15	0.341	16.1	0.12	0.886	18.4	0.42	0.789
Erythrocyte Volume	Time	preT	15.9	0.09	0.912	15.9	2.82	0.089	16.3	3.20	**0.041**
		0	15.8	2.60	0.106	15.8	1.41	0.273	16.8	0.40	0.809
		1	15.9	0.54	0.595	15.9	15.53	**<0.001**	16.3	3.46	**0.032**
		24	10.9	3.37	0.073	8.4	0.18	0.841	9.0	5.87	**0.013**
Red Blood Cell (RBC)	Time	preT	16.1	1.46	0.261	16.1	6.92	**0.007**	16.8	1.24	0.334
		0	16.2	6.11	**0.011**	16.2	2.15	0.149	17.5	2.09	0.126
		1	16.1	15.38	**<0.001**	16.1	4.18	**0.035**	16.9	4.73	**0.010**
		24	16.2	0.65	0.534	16.2	0.24	0.792	17.7	0.17	0.952
White blood cell (WBC)	Time	preT	16.9	2.00	0.166	16.9	1.43	0.267	23.7	2.36	0.083
		0	15.7	11.81	**0.001**	15.7	5.80	**0.013**	21.3	0.83	0.520
		1	16.0	9.51	**0.002**	16.0	2.39	0.123	24.8	0.47	0.756
		24	12.6	6.13	0.014	8.3	15.64	**0.002**	12.2	7.69	**0.003**
Lactate Concentration (LA)	Time	preT	17.8	1.80	0.194	17.8	0.79	0.471	26.8	0.35	0.838
		3postT	14.5	2.04	0.166	8.5	8.61	**0.009**	11.2	0.82	0.536
		20postT	10.0	1.90	0.201	8.4	0.11	0.901	13.5	0.46	0.762

Body mass (ΔBM): between pre-test (preT) and post-test (postT) values during exercise. Measurement times are reported as pre-test and at 0, 1, and 24 h after the exertion test. Lactate concentration (LA) measured before the test (preT) and at 3 and 20 min after the exercise test. Significance (*p*) and (F) statistic with degrees of freedom (DFl—numerator, DFm—denominator) for fixed effects (hydration strategy, test order, and their interaction), statistically significant values have been bolded.

**Table 4 biology-12-00687-t004:** Averaged results of three adjacent heart rate (HR_avg3) measurements taken at 80 min during three successive exercise tests (1, 2, 3) performed using three different hydration strategies (I, W, NH): least square means (LSMs) and standard errors (SEs).

	Test Order	1			2			3		
HR										
	Strategy	I	W	NH	I	W	NH	I	W	NH
80	LSM	146.3	146.0	159.5	150.6	126.3	145.5	133.6	139.7	136.6
	SE	5.1	5.1	5.1	5.3	5.3	5.3	5.7	5.7	5.7

**Table 5 biology-12-00687-t005:** Results for leukocyte count (103/mm^3^) measured 24 h after the test, performed under different hydration strategies (I, W, NH) in three successive tests (1, 2, 3): marginal means (LSMs) and standard errors (SEs) for values after logarithmic transformation (log).

	Test Order	1			2			3		
WBC										
	Strategy	I	W	NH	I	W	NH	I	W	NH
24	LSM	0.905	0.789	0.702	0.706	0.640	0.821	0.824	0.793	0.776
	SE	0.04	0.04	0.04	0.02	0.02	0.02	0.03	0.03	0.03

## Data Availability

All data are included in the manuscript.

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
