# Peer review of "The Influence of Various Hydration Strategies (Isotonic, Water, and No Hydration) on Hematological Indices, Plasma Volume, and Lactate Concentration in Young Men during Prolonged Cycling in Elevated Ambient Temperatures"

_biology, 2023, doi:10.3390/biology12050687_

Round 1

Reviewer 1 Report (Previous Reviewer 1)

Dear Authors,

Manuscript Number: biology-2294097

Title Manuscript: The influence of various hydration strategies on hematological indices, plasma volume, and lactate concentration in young men during prolonged physical exertion in elevated ambient temperatures

I am very grateful to the authors for their efforts.

In general, this manuscript has found suitable content after correcting major revisions, and the modified revisions are accepted.

Best Regards

 8 April 2023

Author Response

Thank you 

Reviewer 2 Report (Previous Reviewer 2)

All my questions have been well addressed, I recommend to accept now. 

Author Response

Thank you 

Reviewer 3 Report (Previous Reviewer 4)

The authors improved the paper and reduced the number of tables. However, now the authors present an extreme great table (Tab. 3) which makes it not easier to understand the results. I still recommend a visualization of the results. As presented the data are hard to understand.

There are stil s some typing and editorial errors. Examples:

ln. 147: ...seconds... 

ln. 177 ff: mixing physical dimensions and parameters, i.e. 'minute oxygen uptake', 'minute ventilation'

ln. 323: 'body weight' (remark: the abreviations should be consequently applied)

ln. 166: What is meant with 'Indicators of exercise respiratory metabolism...'?

ln. 317 ff: The first sentence of this paragraph is superfluid. The other  statements should appear in the methods section.

Author Response

Dear Reviewer,

Thank you very much for your time and valuable comments, which all have been considered and incorporated. The detailed list of responses is given below. We hope that the modifications and explanation will be acceptable for you.

Yours sincerely,

Rydzik, corresponding author

The authors improved the paper and reduced the number of tables. However, now the authors present an extreme great table (Tab. 3) which makes it not easier to understand the results. I still recommend a visualization of the results. As presented the data are hard to understand.

A: Thank you for appreciating our hard work. We have reduced a large number of tables, so Table 3 may seem too long, but upon acceptance of the manuscript after full typesetting, it will be in a collapsible form. However, we have made an effort to improve the readability of the table by bolding statistically significant values, making it easier to interpret. We have also reviewed the entire text and found minor shortcomings which have already been corrected.

There are stil s some typing and editorial errors. Examples:

ln. 147: ...seconds... 

A: This has been corrected

ln. 177 ff: mixing physical dimensions and parameters, i.e. 'minute oxygen uptake', 'minute ventilation'

A: This has been corrected

ln. 323: 'body weight' (remark: the abreviations should be consequently applied)

A: This has been corrected

ln. 166: What is meant with 'Indicators of exercise respiratory metabolism...'?

A: This has been corrected

ln. 317 ff: The first sentence of this paragraph is superfluid. The other  statements should appear in the methods section.

A: This has been corrected

Round 2

Reviewer 3 Report (Previous Reviewer 4)

Only parts of the remarks were tackled. Most critical are missing illustrations of the results. This makes it nearly impossible to follow the discussion. 

In the last review EXAMPLES were listed. some were corrected, some not. E.g. the phrase 'Indicators of exercise respiratory metabolism' was not tackled, 'minute ventilation' etc. still is in the text. For variables derived by time, this should be abbreviated by V or V'.

Even the discussion is correct it is suggested to focuss it to the main findings.

Author Response

Dear Reviewer,

Thank you very much for your time and valuable comments, which all have been considered and incorporated. The detailed list of responses is given below. We hope that the modifications and explanation will be acceptable for you.

Yours sincerely,

Rydzik, corresponding author

Only parts of the remarks were tackled. Most critical are missing illustrations of the results. This makes it nearly impossible to follow the discussion. 

A: Charts have been added making the discussion seem more coherent. Apologies for previous shortcomings

In the last review EXAMPLES were listed. some were corrected, some not. E.g. the phrase 'Indicators of exercise respiratory metabolism' was not tackled, 'minute ventilation' etc. still is in the text. For variables derived by time, this should be abbreviated by V or V'.

A: We have corrected the notation using a V with a dot. We also apologise for the earlier oversight

Even the discussion is correct it is suggested to focuss it to the main findings.

A: We have made changes and corrections throughout the manuscript

Round 3

Reviewer 3 Report (Previous Reviewer 4)

In this version several figures are included representing the the results. Although this reviewer recommend to illustrate the figures it was not meant to show all data. As already stated, the authors should focus on the real important results and to focus the discussion onto those! 

Author Response

Dear Reviewer, 

Thank you again for your comments. We have shortened our manuscript and made changes to the discussion. We hope this will be acceptable to you.

Your Sincerly, 

dr Łukasz Rydzik 

Round 4

Reviewer 3 Report (Previous Reviewer 4)

Even I still see a potential chance to reduce results and method section I don't want to stopp the process if the editor has no concern according to length, no. of figures an tables.

This manuscript is a resubmission of an earlier submission. The following is a list of the peer review reports and author responses from that submission.

Round 1

Reviewer 1 Report

Dear Authors,

Manuscript Number: biology-2294097

Title Manuscript: The influence of various hydration strategies on hematological indices, plasma volume, and lactate concentration in young men during prolonged physical exertion in elevated ambient temperatures

This study examined the impact of different hydration strategies on hematological indicators and lactate concentration during prolonged physical exertion in a high-temperature environment in young men. This study is an interesting topic but at the moment MAJOR REVISIONS are necessary in order to make it suitable for a final decision for “Biology”.

POINTs of STRENGTH:

1) The effect of different hydration strategies on hematological indicators and lactate concentration during prolonged physical exertion in a high-temperature environment in young men;

2) Optimal arrangement of the results section;   

POINTs of WEAKNESS (and/or should be revised to improve the manuscript):

Abstract:

3) Please provide the type of the study as well as mean BMI of participants in the methods section of the abstract;

4) Please remove the references (Mitchell et al., 1994, and Shirreffs et al., 1996) as well as this sentence (The study was conducted at the Laboratory of Physiological Adaptation at the Academy of Physical Education in Krakow) from the methods section of the abstract;

5) Please add the keyword “Young men” to the keywords section;

1. Introduction:

6) The hypothesis and purpose of this study can be stated in more detail;

2. Materials and methods

7) Please remove this wrong and irrelevant content from the “Materials and methods” section [line N. 97-111]; before submitting the article, authors should review their manuscript several times to avoid such common mistakes:

“[The Materials and Methods should be described with sufficient details to allow oth- 97

ers to replicate and build on the published results. Please note that the publication of your 98

manuscript implicates that you must make all materials, data, computer code, and proto- 99

cols associated with the publication available to readers. Please disclose at the submission 100

stage any restrictions on the availability of materials or information. New methods and 101

protocols should be described in detail while well-established methods can be briefly de- 102

scribed and appropriately cited. 103

Research manuscripts reporting large datasets that are deposited in a publicly avail- 104

able database should specify where the data have been deposited and provide the relevant 105

accession numbers. If the accession numbers have not yet been obtained at the time of 106

submission, please state that they will be provided during review. They must be provided 107

prior to publication. 108

Interventionary studies involving animals or humans, and other studies that require 109

ethical approval, must list the authority that provided approval and the corresponding 110

ethical approval code.111].

Research Material

8) The type of study is unclear. Please specify;

9) The recruitment/screening process of participants OR inclusion and exclusion criteria should be described in more detail;

Research Methods

10) Tabela 1 OR Table 1? Please correct (Fig. 1);

Statistical Analysis

11) Did authors use a statistical software to calculate the sample size? IF YES, please add software name and its findings;

3. Results

12) The results section of this study is well written.

4. Discussion and 5. Conclusions

13) Please move and modify this sentence to the first part of the discussion as follows:

“The aim of the study was to investigate the effectiveness of different hydration strategies (carbohydrate-electrolyte beverage, water, or no hydration) during exercise in high ambient temperatures on hematological indices and lactate concentration in the body.” Both domestic and foreign….

14) What does this study add to the literature? Please explain and add in the conclusions section.

15) Please provide Practical perspectives for this manuscript.

References

16) “References” section is not always in accordance with the authors' guidelines. In particular, please check No. 4, 16, 23, 26, and 34 for validation.

 Best Regards

15 March 2023

Author Response

Reviewer 1

Dear Reviewer,

Thank you very much for your time and valuable comments, which all have been considered and incorporated. The detailed list of responses is given below. We hope that the modifications and explanation will be acceptable for you.

Yours sincerely,

Rydzik, corresponding author

Dear Authors,

Manuscript Number: biology-2294097

Title Manuscript: The influence of various hydration strategies on hematological indices, plasma volume, and lactate concentration in young men during prolonged physical exertion in elevated ambient temperatures

This study examined the impact of different hydration strategies on hematological indicators and lactate concentration during prolonged physical exertion in a high-temperature environment in young men. This study is an interesting topic but at the moment MAJOR REVISIONS are necessary in order to make it suitable for a final decision for “Biology”.

POINTs of STRENGTH:

1) The effect of different hydration strategies on hematological indicators and lactate concentration during prolonged physical exertion in a high-temperature environment in young men;

2) Optimal arrangement of the results section;   

POINTs of WEAKNESS (and/or should be revised to improve the manuscript):

Abstract:

3) Please provide the type of the study as well as mean BMI of participants in the methods section of the abstract;

The research method was quasi-experimental. The study involved 12 healthy men aged 20.67 ± 0.98 years and BMI of 23.60± 0.48.

4) Please remove the references (Mitchell et al., 1994, and Shirreffs et al., 1996) as well as this sentence (The study was conducted at the Laboratory of Physiological Adaptation at the Academy of Physical Education in Krakow) from the methods section of the abstract;

Delated.

5) Please add the keyword “Young men” to the keywords section;

Added.

  1. Introduction:

6) The hypothesis and purpose of this study can be stated in more detail;

Maintaining an optimal state of hydration during exercise becomes more complicated depending on the sport, type of activity and availability of fluid. Optimal hydration is dependent on many factors but can generally be defined during exercise as avoiding losses greater than 2–3% of body mass while also avoiding over hydration [1]. Furthermore, during exercise, it is not uncommon for individuals to involuntarily dehydrate, in which they consume less fluid than their fluid needs. Excessive fluid intake can also be problematic, with hyponatremia developing in severe cases of overhydration [1]. Inappropriate management of fluid intake resulting in hypohydration, or hyperhydration, can be detrimental for performance and in some circumstances, increases health risk. The loss of body water during exercise exacerbates physiological and perceptual strain [2]and it is well established that these changes can impair endurance performance, particularly in hot environments and may increase the risk of exertional heat illness [1].

  1. Materials and methods

7) Please remove this wrong and irrelevant content from the “Materials and methods” section [line N. 97-111]; before submitting the article, authors should review their manuscript several times to avoid such common mistakes:

“[The Materials and Methods should be described with sufficient details to allow oth- 97

ers to replicate and build on the published results. Please note that the publication of your 98

manuscript implicates that you must make all materials, data, computer code, and proto- 99

cols associated with the publication available to readers. Please disclose at the submission 100

stage any restrictions on the availability of materials or information. New methods and 101

protocols should be described in detail while well-established methods can be briefly de- 102

scribed and appropriately cited. 103

Research manuscripts reporting large datasets that are deposited in a publicly avail- 104

able database should specify where the data have been deposited and provide the relevant 105

accession numbers. If the accession numbers have not yet been obtained at the time of 106

submission, please state that they will be provided during review. They must be provided 107

prior to publication. 108

Interventionary studies involving animals or humans, and other studies that require 109

ethical approval, must list the authority that provided approval and the corresponding 110

ethical approval code.111].”

 Deleted.

Research Material

8) The type of study is unclear. Please specify;

The research method was quasi-experimental.

9) The recruitment/screening process of participants OR inclusion and exclusion criteria should be described in more detail;

 12 healthy people were selected as available and purposefully.

During the period of the experiment, study participants did not use any stimulants, vitamins, and other supplements. A total of 12 people completed the full cycle of tests.

Research Methods

10) Tabela 1 OR Table 1? Please correct (Fig. 1);

Statistical Analysis

11) Did authors use a statistical software to calculate the sample size? IF YES, please add software name and its findings;

 The SAS 9.3 software (SAS Institute Inc., Cary, NC, USA) was used for the analysis.

  1. Results

12) The results section of this study is well written.

  1. Discussionand 5. Conclusions

13) Please move and modify this sentence to the first part of the discussion as follows:

“The aim of the study was to investigate the effectiveness of different hydration strategies (carbohydrate-electrolyte beverage, water, or no hydration) during exercise in high ambient temperatures on hematological indices and lactate concentration in the body.” Both domestic and foreign….

14) What does this study add to the literature? Please explain and add in the conclusions section.

My own research shows that regardless of the hydration strategy used, experimental exercise caused a decrease in erythrocyte volume, although the changes were small. Significant differences in the effectiveness of different hydration strategies could be observed immediately after exercise. The smallest changes were noted when using an isotonic drink, medium when administering water, and the largest when no hydration was used. Such dependencies were also noted an hour after exercise, but the average ∆MCV differences were small and not statistically significant.

It is worth noting that even 24 hours after the exercise trial, erythrocyte volumes did not return to pre-exercise values regardless of the hydration strategy used. In men who did not hydrate during the test, a clearly higher degree of dehydration of red blood cells was observed at 24 hours than when using an isotonic drink or water, even though other parameters such as plasma volume or body weight returned to pre-test values or even increased.

The use of an isotonic drink probably maintains the proper concentration of osmotically active compounds, and such a state inhibits the transport of water resources from the intracellular space to the outside of the cells, thus preventing their dehydration. It can therefore be assumed that isotonic drinks also effectively eliminate heat from the body, but at the same time do not cause extracellular and intracellular dehydration, as evidenced by the results of changes in erythrocyte volume.

The results obtained confirm that when an isotonic drink is consumed, there are no significant changes in osmolal pressure between the body's water spaces. The volume of plasma did not decrease significantly, and the oncotonic pressure of plasma protein colloid did not increase, thus reducing the dehydration of red blood cells. Therefore, the use of an isotonic drink is more effective than water in maintaining exercise performance, positively affecting blood fluidity and cell elasticity, especially erythrocytes.

15) Please provide Practical perspectives for this manuscript.

The isotonic drink was the most effective way to hydrate the extracellular space, causing the least changes in hematological indicators.

References

16) “References” section is not always in accordance with the authors' guidelines. In particular, please check No. 4, 16, 23, 26, and 34 for validation.

Reviewer 2 Report

This manuscript entitled “The influence of various hydration strategies on hematological indices, plasma volume, and lactate concentration in young men during prolonged physical exertion in elevated ambient temperatures” was primarily aimed to explore the effects of different hydration strategies on hematological indicators and lactate concentration during pro-longed physical exertion in a high-temperature environment in. The authors bring an interesting study, but there are still some problems that cannot up this article to a publishing level. Suggestions are listed in the specific comments below.

Specific comments:

1.     In the abstract part, line 25-26, “The study involved 12 healthy men aged 20.67 ± 0.98 years.” Please provide detailed anthropometry information for participants such as height, weight and body mass index.

2.     For the Abstract part, authors provide too much description, especially the Method. Please simplify it.

3.     In the introduction part, line 67-68, “Available publications indicate that this effectively improves exercise performance [9].” Please cite more relevant papers here to support this statement.

4.     In the introduction part, line 87-89, “It has been shown that an increase in hematocrit (HCT) to 60% can impair blood flow through capillaries, which can have a detrimental effect on the body's exercise capacity.” Please add the relevant paper here.

5.     In the last paragraph of the introduction part, the research gap lacks, which make the novelty and value of this study unclear.

6.     In the Materials and Methods part, the first three paragraphs, line 97-111, I think it is the template of this journal, please check and delete it.

7.     In the Materials and Methods part, Research Material, line 113-115, “The study group consisted of 12 selected healthy males with an average age of 20.67 ± 0.98 years, characterized by an average level of aerobic fitness according to American Heart Association norms (2003).” Please provide detailed anthropometry information for participants such as height, weight.

8.     In the discussion part, it is recommended to provide a brief description of the aim and main findings in the first paragraph of the manuscript.

9.     In the discussion part, line 640-653 and line 654-667, these two parts are absolutely the same. You really should carefully check this manuscript. Some recently studies could be added in the discussion, such as:

The Effect of Acute Aerobic Exercise with Music on Executive Function: The Major Role of Tempo Matching. Physical Activity and Health, 5(1), p.31–44.

The Impact of Aerobic Exercise on HDL Quantity and Quality: A Narrative Review. Int. J. Mol. Sci. 2023, 24, 4653. https://doi.org/10.3390/ijms24054653

10.  What are the limitations of this study? Please provide relevant description.

11.  In the conclusion part, please also provide relevant descriptions about the implications for the future studies.

12.  Please do check the language and grammar mistakes throughout the whole article to further improve clarity.

Author Response

Reviewer 2

Dear Reviewer,

Thank you very much for your time and valuable comments, which all have been considered and incorporated. The detailed list of responses is given below. We hope that the modifications and explanation will be acceptable for you.

Yours sincerely,

Rydzik, corresponding author

This manuscript entitled “The influence of various hydration strategies on hematological indices, plasma volume, and lactate concentration in young men during prolonged physical exertion in elevated ambient temperatures” was primarily aimed to explore the effects of different hydration strategies on hematological indicators and lactate concentration during pro-longed physical exertion in a high-temperature environment in. The authors bring an interesting study, but there are still some problems that cannot up this article to a publishing level. Suggestions are listed in the specific comments below.

Specific comments:

  1. In the abstract part, line 25-26, “The study involved 12 healthy men aged 20.67 ± 0.98 years.” Please provide detailed anthropometry information for participants such as height, weight and body mass index.

The study involved 12 healthy men aged 20.67 ± 0.98 years, were characterized by body height (BH) of 177.25 ± 4.83 cm, body mass (BM) of 74.45 ± 7.6 kg, lean body mass (LBM) of 61.18 ± 6.19 kg .

  1. For the Abstract part, authors provide too much description, especially the Method. Please simplify it.

Done.

Delated:Hydration strategies were based on standards adopted by the American College of Sports Medi-cine (ACSM). The study was conducted at the Laboratory of Physiological Adaptation at the Academy of Physical Education in Krakow

  1. In the introduction part, line 67-68, “Available publications indicate that this effectively improves exercise performance [9].” Please cite more relevant papers here to support this statement.

Added. Kreider, R.B., et al., ISSN exercise & sport nutrition review: research & recommendations. Journal of the international society of sports nutrition, 2010. 7(1): p. 7.

Exercise-related hemoconcentration and hemodilution in hydrated and dehydrated athletes: An observational study of the Hungaria

  1. In the introduction part, line 87-89, “It has been shown that an increase in hematocrit (HCT) to 60% can impair blood flow through capillaries, which can have a detrimental effect on the body's exercise capacity.” Please add the relevant paper here.

Added.Huisjes, R., et al., Squeezing for life–properties of red blood cell deformability. Frontiers in physiology, 2018. 9: p. 656.

Komka, Z.; Szilágyi, B.; Molnár, D.; Sipos, B.; Tóth, M.; Sonkodi, B.; Ács, P.; Elek, J.; Szász, M. Exercise-related hemoconcentration and hemodilution in hydrated and dehydrated athletes: An observational study of the Hungarian canoeists. PLoS One 2022, 17, e0277978, doi:10.1371/journal.pone.0277978.

  1. In the last paragraph of the introduction part, the research gap lacks, which make the novelty and value of this study unclear.

Added: Maintaining an optimal state of hydration during exercise becomes more complicated depending on the sport, type of activity and availability of fluid. Optimal hydration is dependent on many factors but can generally be defined during exercise as avoiding losses greater than 2–3% of body mass while also avoiding over hydration [3]. Furthermore, during exercise, it is not uncommon for individuals to involuntarily dehydrate, in which they consume less fluid than their fluid needs. Excessive fluid intake can also be problematic, with hyponatremia developing in severe cases of overhydration [3]. Inappropriate management of fluid intake resulting in hypohydration, or hyperhydration, can be detrimental for performance and in some circumstances, increases health risk. The loss of body water during exercise exacerbates physiological and perceptual strain [4]and it is well established that these changes can impair endurance performance, particularly in hot environments and may increase the risk of exertional heat illness [3].

  1. In the Materials and Methods part, the first three paragraphs, line 97-111, I think it is the template of this journal, please check and delete it.

Deleted.

  1. In the Materials and Methods part, Research Material, line 113-115, “The study group consisted of 12 selected healthy males with an average age of 20.67 ± 0.98 years, characterized by an average level of aerobic fitness according to American Heart Association norms (2003).” Please provide detailed anthropometry information for participants such as height, weight.

Added:The study involved 12 healthy men aged 20.67 ± 0.98 years, were characterized by body height (BH) of 177.25 ± 4.83 cm, body mass (BM) of 74.45 ± 7.6 kg, lean body mass (LBM) of 61.18 ± 6.19 kg

  1. In the discussion part, it is recommended to provide a brief description of the aim and main findings in the first paragraph of the manuscript.

Added:The aim of the study was to investigate the effectiveness of different hydration strategies (carbohydrate-electrolyte beverage, water, or no hydration) during exercise in high ambient temperatures on hematological indices and lactate concentration in the body. Based on the results of the study, it was shown that hydration with a carbohydrate-electrolyte beverage compared to water was more effective in regulating the permeation of body water between intra- and extracellular spaces, resulting in a smaller loss of serum volume. Significant differences in serum volume were observed between the use of isotonic beverage and no hydration and between the use of isotonic beverage and water. Immediately after the experimental exercise, hemoglobin values were significantly higher with no hydration than with water. An even stronger significance of differences in hemoglobin was observed between no hydration and isotonic beverage consumption. There was a statistically significant difference in the number of leukocytes between the consumption of isotonic beverage and no hydration.

  1. In the discussion part, line 640-653 and line 654-667, these two parts are absolutely the same. You really should carefully check this manuscript. Some recently studies could be added in the discussion, such as:

Corrected.

The Effect of Acute Aerobic Exercise with Music on Executive Function: The Major Role of Tempo Matching. Physical Activity and Health, 5(1), p.31–44.

Added:

The Impact of Aerobic Exercise on HDL Quantity and Quality: A Narrative Review. Int. J. Mol. Sci. 2023, 24, 4653. https://doi.org/10.3390/ijms24054653

Added:

  1. What are the limitations of this study? Please provide relevant description.

Due to the inability to meet clear criteria for reliable measurement, the study did not use bioelectrical impedance analysis to assess body composition at the end of the experiment. Additionally, the study did not investigate ROS and RNS. These limitations should be taken into account when interpreting the results.In the conclusion part, please also provide relevant descriptions about the implications for the future studies.

Practical Implication

   The isotonic drink was the most effective way to hydrate the extracellular space, causing the least changes in hematological indicators.

  1. Please do check the language and grammar mistakes throughout the whole article to further improve clarity.

It was reviewed.

Reviewer 3 Report

General comments

The goal of this study was to examine, as stated by the authors, the impact of different hydration strategies on hematological indicators and lactate concentration during prolonged physical exertion in a high-temperature environment in young men. The authors concluded that each active hydration strategy allows for better maintenance of water-electrolyte homeostasis during physical exertion in a high-temperature environment, and isotonic beverage consumption was found to be the most effective way of hydrating extracellular spaces with the smallest changes in hematological indicators. Overall, the manuscript is weak. First, it brings little new to the literature as several studies have already been conducted on this particular topic. Unfortunately, authors were unable to showcase the novelty of their study in the introduction section. They contended to make a general literature review, without providing a clear, physiologically-based rationale for why there was a need to compare those 3 hydration strategies. Second, authors rehydrated the participants during exercise at a rate corresponding to 120-150% of their sweat rates, which represent a major flaw and problem. Indeed, to this day, sporting organizations no longer recommend fluid replacement at a rate higher than the sweat rate, as it may predispose individuals to hyponatremia, a potentially fatal condition. Hence, in addition of encouraging a non-ethical way of replacing fluid during exercise, results of this study are not ecologically valid. Finally, there was no control for hydration, exercise practice, diet and circadian rhythm before exercise, which all represent major flaws in that it may have confounded findings. My comments are below for authors to consider.    

 Specific comments

Title

Please indicate the 3 strategies compared. Moreover, replace physical exertion by prolonged cycling.

Abstract

Lines 19-22. These are general statements about how heat stress and proper hydration impact physiology. However, it is of little value or interest to the readers of the current paper who want to know why you performed that study. Hence, please provide a clear, physiologically-based rationale for why there was a need to compare those 3 hydration strategies and, as important, what your study will bring to science in terms of novelty.  

Line 24. Please mention the three hydration strategies as well as the hematological parameters. Please do not use the term exertion, replace by exercise, apply throughout.

Line 26. Age, one decimal please.

Line 27. Please indicate whether it was randomized, crossover?

Line 28. Please indicate the intensity?

Line 30. This mode of fluid replacement is totally unethical and can be dangerous. Indeed, you overhydrated participants during exercise and this may lead to hyponatremia, a potentially fatal condition.

Line 31. This is not correct, as the ACSM does not recommend drinking at a rate superior than sweat rate. Please revise.

Lines 33-34. Please remove, not necessary.

Line 34-37. What does that mean? Please show the findings instead!

Line 44. You showed no data to demonstrate that electrolyte homeostasis was better preserved.

Line 45. It is not correct to say that isotonic beverage was found to be most effective. Your results are not in line with that. Relying only on a stronger P value does not allow you to say that.  Both solutions were as effective.

Introduction

The introduction is weak overall. Indeed, it does not provide the rationale for why it is important to compare those 3 hydrations strategies. It does provide the specific, underlying physiological reasons for why those 3 hydration strategies should impact the outcomes differently. Plenty of research has already been conducted on this topic. However, the novelty of the study has not been indicated and contrasted against the current knowledge on the topic.  

Line 97-111. Please remove.

Methods

Figure 1. First week of research should be at the top of the figure, not bottom. Please, flip over.

Line 139. Why a Wingate test? Why was it done in relation to the current goals of the study? I imagine that this study represents secondary analyses of a larger project. It needs to be mentioned beforehand, and not come out of the blue, as is the case at the moment.

Line 176 and 185. Wind speed was much too low. This represents a major problem with the study, as convective cooling was prevented, which is a major route for heat loss during cycling during out-of-doors conditions.

Lines 207. This comes out of the blue. Why was it done? Please state beforehand what is the true purpose of the study. I imagine there are several. Please report those numbers in the abstract, also.

Results

This section as highly problematic. Too many results and details are reported. Readers will get easily lost and bored by the section. It seems as though the results from a master's thesis or doctoral dissertation have been reported in full with no attempts to discriminate the important from the more futile results. This section needs to be completely rearranged. 21 tables are reported in this section, which is nonsense. Moreover, several results would benefit from being reported as figures. I encourage authors to have a look at papers published on this topic to get an idea of the ideal number of figures and tables that is adequate.

Author Response

Reviewer 3

Dear Reviewer,

Thank you very much for your time and valuable comments, which all have been considered and incorporated. The detailed list of responses is given below. We hope that the modifications and explanation will be acceptable for you.

Yours sincerely,

Rydzik, corresponding author

General comments

The goal of this study was to examine, as stated by the authors, the impact of different hydration strategies on hematological indicators and lactate concentration during prolonged physical exertion in a high-temperature environment in young men. The authors concluded that each active hydration strategy allows for better maintenance of water-electrolyte homeostasis during physical exertion in a high-temperature environment, and isotonic beverage consumption was found to be the most effective way of hydrating extracellular spaces with the smallest changes in hematological indicators. Overall, the manuscript is weak. First, it brings little new to the literature as several studies have already been conducted on this particular topic. Unfortunately, authors were unable to showcase the novelty of their study in the introduction section. They contended to make a general literature review, without providing a clear, physiologically-based rationale for why there was a need to compare those 3 hydration strategies. Second, authors rehydrated the participants during exercise at a rate corresponding to 120-150% of their sweat rates, which represent a major flaw and problem. Indeed, to this day, sporting organizations no longer recommend fluid replacement at a rate higher than the sweat rate, as it may predispose individuals to hyponatremia, a potentially fatal condition. Hence, in addition of encouraging a non-ethical way of replacing fluid during exercise, results of this study are not ecologically valid. Finally, there was no control for hydration, exercise practice, diet and circadian rhythm before exercise, which all represent major flaws in that it may have confounded findings. My comments are below for authors to consider.    

 Specific comments

Title

Please indicate the 3 strategies compared. Moreover, replace physical exertion by prolonged cycling.

The influence of various hydration strategies (Isotonic, Water and No Hydration) on hematological indices, plasma volume, and lactate concentration in young men during prolonged cyclingin elevated ambient temperatures

Abstract

Lines 19-22. These are general statements about how heat stress and proper hydration impact physiology. However, it is of little value or interest to the readers of the current paper who want to know why you performed that study. Hence, please provide a clear, physiologically-based rationale for why there was a need to compare those 3 hydration strategies and, as important, what your study will bring to science in terms of novelty.  

An appropriate drinking strategy will take account of pre-exercise hydration status and of fluid,electrolyte and substrate needs before, during and after exercise.

Line 24. Please mention the three hydration strategies as well as the hematological parameters. Please do not use the term exertion, replace by exercise, apply throughout.

The objective of this study was to assess the impact of different hydration strategies (Isotonic, Water and No Hydration) on hematological indicators (hemoglobin concentration, hematocrit number, erythrocyte count, leukocyte count, mean corpuscular volume, and platelet count) and lactate concentration during prolonged physical exertion in a high-temperature environment in young men.

Line 26. Age, one decimal please

Done.

Line 27. Please indicate whether it was randomized, crossover?

Done.

Line 28. Please indicate the intensity?

intensity of 110 W

Line 30. This mode of fluid replacement is totally unethical and can be dangerous. Indeed, you overhydrated participants during exercise and this may lead to hyponatremia, a potentially fatal condition.

Line 31. This is not correct, as the ACSM does not recommend drinking at a rate superior than sweat rate. Please revise.

Deleted.

Lines 33-34. Please remove, not necessary.

Deleted

Line 34-37. What does that mean? Please show the findings instead!

Deleted

Line 44. You showed no data to demonstrate that electrolyte homeostasis was better preserved.

Line 45. It is not correct to say that isotonic beverage was found to be most effective. Your results are not in line with that. Relying only on a stronger P value does not allow you to say that.  Both solutions were as effective.

Corrected.

Introduction

The introduction is weak overall. Indeed, it does not provide the rationale for why it is important to compare those 3 hydrations strategies. It does provide the specific, underlying physiological reasons for why those 3 hydration strategies should impact the outcomes differently. Plenty of research has already been conducted on this topic. However, the novelty of the study has not been indicated and contrasted against the current knowledge on the topic.  

Line 97-111. Please remove.

Deleted.

Methods

Figure 1. First week of research should be at the top of the figure, not bottom. Please, flip over.

Line 139. Why a Wingate test? Why was it done in relation to the current goals of the study? I imagine that this study represents secondary analyses of a larger project. It needs to be mentioned beforehand, and not come out of the blue, as is the case at the moment.

Line 176 and 185. Wind speed was much too low. This represents a major problem with the study, as convective cooling was prevented, which is a major route for heat loss during cycling during out-of-doors conditions.

Lines 207. This comes out of the blue. Why was it done? Please state beforehand what is the true purpose of the study. I imagine there are several. Please report those numbers in the abstract, also.

Results

This section as highly problematic. Too many results and details are reported. Readers will get easily lost and bored by the section. It seems as though the results from a master's thesis or doctoral dissertation have been reported in full with no attempts to discriminate the important from the more futile results. This section needs to be completely rearranged. 21 tables are reported in this section, which is nonsense. Moreover, several results would benefit from being reported as figures. I encourage authors to have a look at papers published on this topic to get an idea of the ideal number of figures and tables that is adequate.

Reviewer 4 Report

The authors collected interesting data for an actual topic. However, the presentation of the data is too abstract. This reviewer strongly recommends a presentation in terms of graphical presentation. 21 tables are not acceptable!

Details:

-        Fig 1. is hard to understand.

-        Ln. 163: There was no analysis of kinetics performed.

-        Ln.. 168:  ‘Minute oxygen uptake’ (similar for CO2-output) is an uncommon term and not in line with the use of parameter (mixture of unit and parameter).
For exchange data use ‘V’ (V̇)or “ V' ” because these are derivates by time.

-        Ln. 224: The abbreviation ‘NH’ is mentioned as ‘BN’ in the text.

-        Ln. 287 ff: It is suggested to ignore the Group as factor in the ANCOVA. With n=2 any effect is critical if statistical significant. For this study influences can be excluded by the randomized design.

-        Ln. 321 ff/Table 2: The information is redundant.

-        Ln. 350 ff: The authors should critically check whether  factors like 'time of measurement' (Tabs. 6ff) should be a factor in the ANCOVA

-        Ln. 640: is there only one author (‘My own research…’)?

Author Response

Reviewer 4

Dear Reviewer,

Thank you very much for your time and valuable comments, which all have been considered and incorporated. The detailed list of responses is given below. We hope that the modifications and explanation will be acceptable for you.

Yours sincerely,

Rydzik, corresponding author

The authors collected interesting data for an actual topic. However, the presentation of the data is too abstract. This reviewer strongly recommends a presentation in terms of graphical presentation. 21 tables are not acceptable!

Details:

-        Fig 1. is hard to understand.

-        Ln. 163: There was no analysis of kinetics performed.

Deleted.

-        Ln.. 168:  ‘Minute oxygen uptake’ (similar for CO2-output) is an uncommon term and not in line with the use of parameter (mixture of unit and parameter).
For exchange data use ‘V’ (V̇)or “ V' ” because these are derivates by time.

-        Ln. 224: The abbreviation ‘NH’ is mentioned as ‘BN’ in the text.

Corrected.

-        Ln. 287 ff: It is suggested to ignore the Group as factor in the ANCOVA. With n=2 any effect is critical if statistical significant. For this study influences can be excluded by the randomized design.

-        Ln. 321 ff/Table 2: The information is redundant.

-        Ln. 350 ff: The authors should critically check whether  factors like 'time of measurement' (Tabs. 6ff) should be a factor in the ANCOVA

-        Ln. 640: is there only one author (‘My own research…’)?

Corrected